# Comparative Analysis of Commercially Available Flavor Oil Sausages and Smoked Sausages

**DOI:** 10.3390/molecules29163772

**Published:** 2024-08-09

**Authors:** Penghui Zhao, Yongqiang An, Zijie Dong, Xiaoxue Sun, Wanli Zhang, Heng Wang, Bing Yang, Jing Yan, Bing Fang, Fazheng Ren, Lishui Chen

**Affiliations:** 1Food Laboratory of Zhong Yuan, Luohe 462300, China; 2Engineering Research Center for Industrial Microbial Resources Development and Application of Henan Province, Luohe 462300, China; 3Key Laboratory of Precision Nutrition and Food Quality, Department of Nutrition and Health, China Agricultural University, Beijing 100193, China

**Keywords:** sausage, volatile flavor substances, texture, gas chromatography-ion mobility spectrometry, processing

## Abstract

This study utilized gas chromatography-ion mobility spectrometry (GC-IMS) to analyze the volatile flavor compounds present in various commercially available sausages. Additionally, it conducted a comparative assessment of the distinctions among different samples by integrating sensory evaluation with textural and physicochemical parameters. The results of the GC-IMS analysis showed that a total of 65 volatile compounds were detected in the four samples, including 12 hydrocarbons, 11 alcohols, 10 ketones, 9 aldehydes, 12 esters, and 1 acids. Fingerprinting combined with principal component analysis (PCA) showed that the volatiles of different brands of sausages were significantly different (*p* < 0.05). The volatiles of S1 and S4 were more similar and significantly different from the other two samples (*p* < 0.05). Among them, there were 14 key volatile substances in the four samples, of which 3-hydroxy-2-butanone and diallyl sulfide were common to all four sausages. Combined textural and sensory evaluations revealed that smoked sausages exhibited superior characteristics in resilience, cohesiveness, springiness, gumminess, and chewiness. Additionally, smoked sausages were found to be more attractive in color, moderately spicy, and salty, while having a lower fat content. In conclusion, smoked sausages are preferred by consumers over flavored oil sausages.

## 1. Introduction

In recent years, the increasing demand for healthy, nutritious, and safe meat products has driven continuous innovation in the food field, and there is an urgent need to develop personalized and functional products that are healthier and more nutritious to meet the needs of the majority of consumers [1]. Sausage is a processed meat product with a strong flavor and unique taste, that has a long history in China [2]. However, there is a wide variety of sausages in the market at present, with varying quality, taste, texture, color, and flavor [3], mainly due to the differences in the preparation of raw materials and the processing techniques used (use of fermentation agents), ingredients (salt type, etc.), and the regulation of environmental conditions (such as temperature, surface molds, smoking, etc.)during the ripening process [4]. Clarifying the link between key processing technologies and product quality is important for sausage quality improvement, and VOCs and texture are considered important indicators in the evaluation of sausage quality [5]. The generation of aroma mainly occurs in the cooking process of sausage, sausage production in some processing processes such as steaming and smoking under the production of aroma, color volatile organic compounds, and lipid metabolites, giving the sausage a unique color, texture, and aroma. The texture of the sausage is also changed during chopping, tumbling, and cooking. This is accompanied by protein denaturation, fat oxidation, and protein-starch cross-linking reactions [6,7,8,9]. Therefore, it is of practical importance to analyze and compare them. Previous studies have done a lot of work on improving the texture and flavor of sausages. It has been shown that adding the addition of probiotics can increase the moisture content and water activity of sausages, reduce their pH value, and produce low-salt sausages with better flavor and texture [10]. Moreover, the addition of Lycium barbarum alters the hardness of the sausage samples with better organoleptic and structural-chemical properties, and its antimicrobial properties also help to prolong the shelf life of the sausages [11]. In recent years, salt-reduced and low-fat products have become a major consumer trend, which has triggered a great deal of attention from researchers. Studies have shown that pretreatment with linseed oil instead of animal fat can significantly improve the nutritional characteristics of sausage [12]. Hu et al. [13] used silverbeet to prepare sausage by substituting fat, and the protein, and ash, and it significantly increased the moisture content, brightness, and redness of the product (*p* < 0.05). The textural and sensory qualities of the sausage were also improved (*p* < 0.05). In addition, the contents of essential and non-essential amino acids in sausages were significantly increased (*p* < 0.05). Studies such as these are increasing, and although researchers and producers continue to reform and innovate in pursuit of the nutritional health of sausages, consumers seem to have higher nutritional quality and health needs. Therefore, there is an urgent need to deepen the processing of sausages and develop high-value sausage products with excellent flavor to meet market demand.

Gas Chromatography-Ion Mobility Spectrometry (GC-IMS) is an emerging technology for the detection of Volatile Organic Compounds (VOCs), which has been widely used in the food industry due to its high efficiency and sensitivity. widely used in the food industry due to its high efficiency and sensitivity [14]. Scientific sensory evaluation plays a key role in the development of novel meat products and is usually used as a basis for product formulation design, and optimization [15]. Combining GC-IMS with correlation analysis of sensory evaluation results helps provide a reliable result. The aim of this study was to conduct a comparative analysis of the flavor substances of different types of sausages using GC-IMS technology, and to further explore the results in combination with the indexes of texture, spiciness, and numbness, to provide a reference for quality assessment, production process improvement, and standardized production of sausages.

## 2. Results and Analysis

### 2.1. Texture Analysis

The Texture Profile Analysis (TPA) tests were conducted on sausages using a texture tester (CTX) to evaluate the textural properties of the four types of sausages concerning five aspects: resilience, cohesiveness, springiness, gumminess, and chewiness (Appendix A). Overall, the cohesiveness, springiness, gumminess, and chewiness parameters of smoked sausages S2 and S3 were significantly higher than those of flavor oil sausages S1 and S4 (*p* < 0.05). Among them, smoked sausage S2 exhibited significantly higher gumminess and chewiness than S3 (*p* < 0.05), while no significant differences were observed in other parameters. Flavor oil sausages S1 and S4 showed a significant difference only in springiness, while the other aspects displayed more or less similar results with no significant difference (*p* < 0.05). Sausage S2 displayed higher index parameters in all aspects. However, based on previous experience, the relative highs and lows of the parameters do not directly indicate quality. Therefore, it is necessary to further explore and summarize the results in combination with sensory evaluation results as a reference standard for subsequent sausage processing (Appendix A).

### 2.2. Physical and Chemical Index Analysis

The fat content plays a pivotal role in influencing the flavor and texture of meat products [16]. High-temperature cooking and baking can induce the oxidation of lipids and proteins, resulting in alterations in protein digestibility and fatty acid composition. These changes impact the flavor and structural characteristics of sausages [17]. Hydroxy-α-sorcinol, being a prominent flavor constituent of peppercorns, is commonly employed as a gauge of pungency in food flavor assessment [18]. Salt, an important food additive, has a significant effect on the processes of fat oxidation, proteolysis, and denaturation. The quantity and method of its addition can alsoimpact the overall quality and flavor of meat products. Additionally, salt can regulate the hydration capacity of proteins, starches, and fats, thereby affecting texture. Furthermore, it can alter the content of water-soluble substances and free fatty acids within the matrix, ultimately impacting flavor [19]. Sucrose is widely used in the food sector as a benchmark substance for assessing the degree of sweetness of sweeteners. However, there are significant differences among different sweeteners [20]. The results of the physicochemical parameters of different sausages are shown in Appendix A. The fat content, spiciness, sodium chloride content, and sucrose content of flavor oil sausages S1 and S4 were higher than those of smoked sausages S2 and S3, indicating that flavor oil sausages possessed sweeter, saltier, and spicier profiles than smoked sausages. However, due to the higher fat content, flavor oil sausages are not recommended for individuals seeking to reduce fat intake. Compared to S4, preserved sausage S1 exhibited higher spiciness and sweetness levels and relatively lower fat and sodium chloride content. Therefore, the sweet and spicy flavors of S1 should be more prominent, while having lower fat and salt content. Overall, the amount of additives used in smoked sausages was lower. The fat content of flavor oil sausages was higher than that of smoked sausages, which could be due to the differences in raw meat parts and processing methods (Appendix A).

### 2.3. GC-IMS Analysis

Figure 1 shows the three-dimensional spectrum of GC-IMS, with three axes indicating migration time (X-axis), retention time (Y-axis) and signal peak intensity (Z-axis), from which the differences in VOCs in different samples can be visualized. For easy observation, the top view is taken below for comparison, as shown in Figure 2.

Description: (1) The background of the whole graph is blue, and the red vertical line at horizontal coordinate 1.0 is the RIP peak (reactive ion peak, normalized). (2) The vertical coordinate represents the retention time (s) of the gas chromatogram, and the horizontal coordinate represents the relative migration time (normalized, a.u.). (3) Each point on either side of the RIP peak represents a VOC. The color represents the peak intensity of the substance, from blue to red, with darker colors indicating greater peak intensity.

To further visualize and compare the differences of volatile components in sausages, the spectrum of the S1 sample was selected as a reference, and the spectra of other samples were deducted from the reference to obtain the comparison between differences of different samples, as shown in Figure 3. If the VOC contents in the target sample and the reference are the same, the background after deduction will be white. The red color shows that the concentration of the substance is higher than the reference in the target sample, while the blue color shows that the concentration of the substance is lower than the reference in the target sample.

The differences in volatile fractions between several groups of samples can be visualized very well in Figure 4. Ethyl butyrate, ethyl propionate, ethyl caproate, 3-hydroxy-2-butanone, 2-pentylfuran, and 3-methylbutanol were the highest in S1. Terpenes such as α-pinene, β-laurene, α-watercressene, α-siderophane, and β-pinene were the highest in S2. Valeraldehyde, 5-methylfurfural, heptanal, furfural, and E-2-methyl-2-butenal were the highest in S3. Ethyl 3-methylbutyrate, ethyl 2-methylbutyrate, ethyl acetate, and furfuryl mercaptan were the highest in S4. The VOC compositions of the four sausages varied markedly, with relative proximity in terms of the VOC compositions and some differences in the contents of S1 and S4.

Description: (1) Each row of the graph represents all the signal peaks selected in one sample. (2) Each column in the graph represents the signal peaks of the same VOC in different samples. (3) The graph shows the complete VOC information for each sample and the differences in VOCs between samples. (4) The peak heights and volumes of all the substances in each sample are shown in an Excel table, which can be imported into a three-way software for statistical analysis. The results of the statistical analysis are based on peak heights.

#### 2.3.1. Volatile Flavor Components in Four Types of Sausages

The relative contents of each type of volatile organic compounds (VOCs) in red oil and smoked sausages are listed in Table 1. A total of 65 VOCs were detected in the four sausages, including 12 hydrocarbons, 11 alcohols, 10 ketones, 9 aldehydes, 12 esters, 1 acids, and 10 others. The relative contents of alcoholic, ester and acid volatiles in flavor oil sausages S1 and S4 were significantly higher than those in smoked sausages S2 and S3 (*p* < 0.05), while the relative contents of ketones and aldehydes in smoked sausages were significantly higher than those in flavor oil sausages (*p* < 0.05).

As depicted in Table 1 and Table 2, the four selected samples each contained seven types of volatile components, totaling 73 volatile flavor substances. Among these, sample S1 exhibited higher relative contents of alcohols, ketones, and acids; sample S2 showed the highest relative contents of hydrocarbons, followed by aldehydes and ketones; sample S3 displayed higher relative contents of ketones and aldehydes, with ketones being the highest; whereas sample S4 demonstrated the most abundant relative contents of ketones and esters, with esters being the highest. Alcohols typically result from fat oxidation and impart relatively softer flavors with botanical and earthy notes. Alcohols and ketones, which were the most abundant and relatively high in the four sausage samples, exert significant influence on the flavor quality of the product and may be introduced through the raw materials. Esters are commonly found in nature, with many exhibiting floral, fruity, and wine-like aromas. Esters in sausages primarily originate from spices and the esterification process, which involves free fatty acids in the meat [21]. High-temperature and high-pressure processing result in elevated levels of esters and aldehydes in the meat. As the duration of high-temperature treatment increases, there is a reduction in meat hardness, rendering the meat tender. Additionally, the chewiness exhibits a pattern of initially decreasing and then increasing over a specific timeframe [22]. Despite the conventional methods employed by sausage manufacturers for processing, variations in baking temperatures and durations among different producers may account for discrepancies in the types and quantities of ester compounds present. The principal origin of aldehydes lies in the oxidation of unsaturated fatty acids. These compounds exhibit a low flavor threshold, contributing distinct sensory profiles characterized by woody, smoky, and burnt aromas, along with fatty flavors. They play a crucial role as constituents of meat flavor [23]. Ketones typically result from reactions such as the Maillard reaction and fat degradation, imparting floral and fruity odors, which are integral components of meat flavor [24]. In sample S3, the highest relative content of ketones was found to be 34.33%, including cyclopentanone, 2-heptanone, 2-acetone, and 2-butanone, which imparted a fruity flavor. The elevated relative content of ketones in these samples significantly contributed to the sausage flavor profile. Hydrocarbon compounds influence meat flavor through various mechanisms, such as fat melting, the smoking process, the Maillard reaction, the release of volatile organic compounds, and the use of seasonings and spices. Generally, hydrocarbon compounds exhibit a high flavor threshold [25]. Acidic compounds primarily arise from the reaction of amino acid degradation products or the degradation of fatty acids, imparting distinctive flavor profiles to foods. In sample S1, the relative content of acid compounds was the highest, reaching 23.09%, with acetic acid being the predominant component, contributing a sour vinegar flavor. In the remaining three samples, the relative content of acetic acid was lower, potentially due to variations in the types and quantities of additives, such as marinades and spices [26]. In sample S2, among other classes of compounds, dipropyl disulfide, 2-methyl-3-keto tetrahydrofuran, and diallyl sulfide exhibited higher relative contents of 0.68%, 1.2%, and 1.45%, respectively, imparting a creamy almond aroma.

Each volatile flavor substance was analyzed using the ROAV method. When a volatile flavor substance of ROAV ≥ 1 is considered to directly impact on the overall flavor, it is regarded as the key volatile flavor substance of the food In the four sausages studied, the ROAV values of the volatile flavor substances are listed in Table 3, except for seven volatile flavor substances for which there is no accessible perception threshold. The volatile flavor substances with ROAVs greater than 1 in the four sample groups were isovaleraldehyde (apple aroma, chocolate aroma), caffealdehyde (pungent odor, similar to garlic or rotten egg), crotonaldehyde (splashy, pungent odor), 2-heptanol, 2-butanone, 3-hydroxy (butterscotch flavor, milk aroma), allyl isothiocyanate (strong, pungent wasabi and horseradish-like odor), ethyl caproate-M (fruity pineapple odor, slight irritating astringency), ethyl butyrate (banana aroma, pineapple-like aroma), ethyl 2-methylpropionate (fruity odor), ethyl propionate (special pineapple and rum aroma), ethyl acetate (etheric, fruity aroma of pineapple, grapes, and cherries), dipropyl disulphide (sulphurous garlic or onion aroma), diallyl sulphide (sulphurous, onion, garlic flavors), for a total of 14. The volatile flavor substances with ROAV greater than 1 in all four groups were 3-hydroxy-2-butanone (ethyl diphosgene) and diallyl sulfide, suggesting that the two are the key flavor components in the four sausages. 2-Butanone, 3-hydroxy is a valuable volatile compound commonly occurring in nature and extensively utilized across various industries, including food, cosmetics, agriculture, and chemicals. It serves as a crucial precursor for synthesizing 2,3-butanediol, liquid hydrocarbon fuels, and heterocyclic compounds [27], and it is commonly used in the food industry as a food additive to enhance the flavor of substrates with a pleasant sour cream and fatty butter flavor [28]. While ethyl diphosgene finds widespread use in food flavoring, it is crucial to acknowledge its inherent toxicity, compounded by the potentially heightened toxicity of its oxidation by-products [29], Some reports have employed it as a testing agent to assess the presence of detrimental bacteria in food samples [30]. This volatile compound exhibits a widespread distribution in nature, with certain microorganisms, higher plants, insects, and higher animals capable of synthesizing 2-Butanone, 3-hydroxy under specific conditions via distinct enzymatic pathways. In contrast, diallyl sulfide primarily originates from garlic and exhibits a range of biological activities, notably antioxidant properties. Its principal mode of action involves the interaction of organosulfides with reactive oxygen species, thereby mitigating oxidative stress [31,32].

#### 2.3.2. Results of Principal Component Analysis of Volatile Flavor Components in Four Types of Sausages

Principal component analysis (PCA) is a method of dimensionality reduction for multivariate, unsupervised learning, also known as principal component analysis. The PCA plot provides a broad overview of the statistical variances among samples within each group and the degree of alignment among samples within the group [33]. In the quest to explore disparities in sausage flavor substances, the peak heights of all identified VOC signals underwent PCA downscaling. The principal component analysis of volatile flavor substances in the four sausage samples was executed through SPSS 23.0 software, as depicted in Figure 5. This analysis considered various sample types as variables. The cumulative contribution of PC1 (45.0%) and PC2 (32.7%) reached 77.7%, indicating the efficacy of these principal components in evaluating the flavor quality across different brands of sausage samples. Tight clustering between parallel samples was observed, indicating good parallelism. Meanwhile, the separation between the samples was high, showing significant intergroup differences. The PCA results reflected differences between the samples, aligning with the visual observation of the fingerprint profiles. Samples S1 and S4 exhibited closer proximity to each other, whereas S2 and S3 demonstrated greater distance from S1 and S4, suggesting significant disparities in aroma compositions possibly attributable to variations in ingredients and processes.

### 2.4. Sensory Analysis

Appendix A displays the sensory scores of various flavor oil sausages and smoked sausages. Overall, smoked sausages S2 and S3 received significantly higher scores than flavor oil sausages S1 and S4 across flavor, organization, color, texture, and overall preference, indicating a preference for smoked sausages likely attributed to variations in processing techniques and additives employed. A comparison of the ratings for smoked sausages S2 and S3 indicated that S3 received slightly higher scores than S2, though the disparity between the two was not statistically significant. Research has demonstrated that the smoking process, including factors such as smoking duration and temperature, impacts textural attributes such as hardness, chewiness, and elasticity of sausages [34]. Furthermore, the smoking process imparts distinctive flavor and appealing color to the sausage, contributing positively to the product’s shelf life [35,36]. Therefore, the sensory scores for S2 and S3 surpassed those of S1 and S4 regarding condition, color, and flavor, aligning with earlier findings from flavor substance determination and textural analysis.

## 3. Materials and Methods

### 3.1. Materials and Reagents

Four distinct sausage samples (commercially available) used were as follows: S1 (originating from Banan, Chongqing, China), S2 (originating from Quanzhou, China), S3 (originating from Luohe, China), and S4 (originating from Zunyi, China).

Methylketones used include the following: 2-butanone, 2-pentanone, 2-hexanone, 2-heptanone, 2-octanone, and 2-nonanone (all analytically pure), Aladdin. Other materials are 99.999% nitrogen; 20 mL headspace vial, Shandong Haineng Scientific Instrument Co., Ltd. (Dezhou, China); and MXT-WAX capillary chromatography column (15 m × 0.53 mm, 1.0 μm), Restek, Inc. (Beijing, China).

### 3.2. Instruments and Equipment

ME20 Electronic Analytical Balance: METTLER-TOLEDO INSTRUMENTS (Shanghai, China) Co., Ltd.; Biochrom30 Amino Acid Auto Analyzer: Biochrom, UK; pH-star MATTHAUS; HR83 Moisture Content Meter METTLER-TOLEDO; XHF-D High-Speed Disperser Ningbo Xinzhi Bio-Chem (Ningbo, China) Co., Ltd.; UV1810S UV Spectrophotometer Shanghai Youke Instrument (Shanghai, China) Co., Ltd.; Colorimeter CR-400 CHROMA METER; FlavourSpec^®^ Gas Phase Ion Mobility Spectrometer, G.A.S. (Dortmund, Germany); CTC-PAL3 Static Headspace Autosampling Device, CTC Analytics AG (Zwingen, Switzerland); VOCOCA (Zwingen, Switzerland); and VOCal data processing software (0.4.03), G.A.S. (Dortmund, Germany).

### 3.3. Experimental Methodology

#### 3.3.1. Measurement of Texture

Sausage texture was determined by the method in reference [13]. A 3 cm × 3 cm × 1.5 cm sausage slice was taken, fixed on the base of the mass structure apparatus, and tested using a P/36R probe. The test parameters were as follows: pre-test rate of 2.5 mm/s, test rate of 1.5 mm/s, post-test rate of 1.0 mm/s, compression ratio of 50%, and trigger force of 4 N. The test parameters were as follows.

#### 3.3.2. Measurement of Physical and Chemical Indicators

The recommended national standards were used to determine the samples’ numbness [37], spiciness [38], salinity [39], sweetness [40] and fat content [41].

#### 3.3.3. Measurement of Volatile Components

The methodology of references [42,43] with modifications was used. The types and relative contents of volatile compounds in sausages were determined by GC-IMS. Sample treatment of 2 g of the sample was accurately weighed into a 20 mL headspace vial and incubated at 60 °C for 15 min. Then the sample was injected into the vial, and three parallel groups were determined for each sample. Headspace injection conditions: incubation temperature: 60 °C; incubation time: 15 min; injection volume: 500 µL; non-shunt injection; incubation speed: 500 r/min; and injection needle temperature: 85 °C

GC conditions: column temperature: 60 °C; carrier gas: high purity nitrogen (purity ≥ 99.999%); and programmed ramping up: The initial flow rate of 2.0 mL/min was maintained for 2 min, and linearly increased to 10.0 mL/min within 8 min. The initial flow rate of 2.0 mL/min was maintained for 2 min. Subsequently, it was incrementally raised to 10.0 mL/min over an 8-min period. Following this, there was a linear increase to 100.0 mL/min within 10 min, which was then sustained for 10 min. The total chromatographic running time amounted to 30 min.

IMS conditions: Ionization source: Tritium source (^3^H); Transfer tube length: 98 mm; Electric field strength: 500 V/cm; Transfer tube temperature: 45 °C; Drift gas: High-purity nitrogen (purity ≥ 99.999%), flow rate: 150.0 mL/min; and Ion mode: Positive ion.

Data processing: A mixture of six ketones was detected to establish calibration curves for retention time and retention index. Subsequently, the retention index of each target compound was calculated based on its retention time. The VOCal software was utilized to retrieve and compare the GC retention indices (NIST 2020) and IMS migration time databases for qualitative analysis of the target compounds.

The VOCal data processing software was employed to generate three-dimensional spectra, two-dimensional spectra, differential spectra, fingerprint spectra, and PCA plots of volatile components using plugins such as Reporter, Gallery Plot, and Dynamic PCA, facilitating comparisons of volatile organic compounds among samples.

#### 3.3.4. Sensory Evaluation

The sensory evaluation was carried out in the Zhongyuan Food Laboratory, located in Luohe, China. The sensory evaluation team comprised 20 members, consisting of 10 females and 10 males aged between 25 and 45 years, all with backgrounds in food science-related disciplines. All team members underwent specialized training in sensory evaluation. The sensory evaluation criteria were based on existing literature methods with modifications [19], focusing on assessing the color, texture, taste, and flavor of the sausage. Percentage scoring criteria were utilized, and the evaluation criteria along with their respective scores are presented in Appendix A.

### 3.4. Statistical Analysis

The data processing was conducted using Excel 2021 and SPSS Statistics 26.0, while data visualization and analysis were performed using Origin 2021 and R 4.1.3. The experimental results were presented as “mean ± standard deviation”, and the significance of the differences was assessed using the Duncan test within a one-way analysis of variance (ANOVA), with a significant threshold of *p* < 0.05 indicating statistically significance. To ensure reliability, all experimental results were measured three times concurrently.

## 4. Conclusions

To investigate variations in taste and volatile flavor compounds across different sausage types, a thorough examination encompassing fundamental physicochemical parameters, texture, structure, sensory evaluation, and GC-IMS analysis of volatile substances was conducted. The GC-IMS analysis revealed the presence of 73 volatile compounds, predominantly comprising alcohols, ketones, acids, esters, hydrocarbons, aldehydes, and other compounds. Notably, alcohols, ketones, and esters exhibited greater abundance both in terms of variety and concentration among the volatile compounds identified. In contrast, acids and other volatile compounds were characterized by fewer types and relatively lower concentrations. The distinctive physicochemical properties and unique aromas exhibited by various sausage brands stem from variations in raw materials, processing techniques, and the synergistic interaction of diverse flavor components. The results of principal component analysis showed that the main flavor substances in the four sausages were isovaleraldehyde, caffealdehyde, crotonaldehyde, 2-heptanol, 2-butanone,3-hydroxy, allyl isothiocyanate, ethyl caproate-M, ethyl butyrate, ethyl 2-methylpropanoate, ethyl propanoate, ethyl acetate, dipropyl disulfide, and diallyl sulfide, a total of 14 kinds of substances, which constituted the main flavors of the sausages. Combining the results of textural and sensory evaluations, we found that smoked sausages performed better than flavor oil sausages in terms of resilience, cohesion, elasticity, viscosity, and chewiness, which was consistent with the results of sensory evaluation. In addition to possessing a more appealing color, moderate levels of spiciness and saltiness, and lower fat content, smoked sausages are generally preferred by consumers over flavor oil sausages. Conducting a comprehensive comparative analysis of their production processes and optimizing key techniques and formulations is essential. The findings of such research endeavors will offer scientific insights and innovative strategies for the development of new flavored sausages and the refinement of traditional meat product processes. This, in turn, will facilitate the transformation and advancement of the food industry, bolstering the market competitiveness of relevant enterprises. Therefore, further exploration and application of related research are warranted to drive continued progress and innovation in this domain.

## Figures and Tables

**Figure 1 molecules-29-03772-f001:**
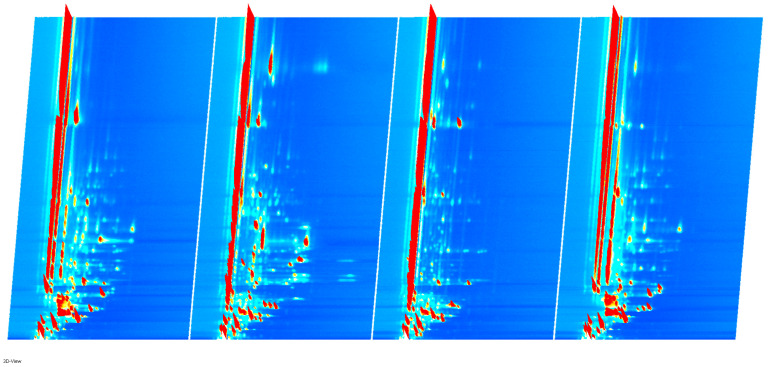
Three-dimensional GC-IMS spectrum of volatile components in sausages.

**Figure 2 molecules-29-03772-f002:**
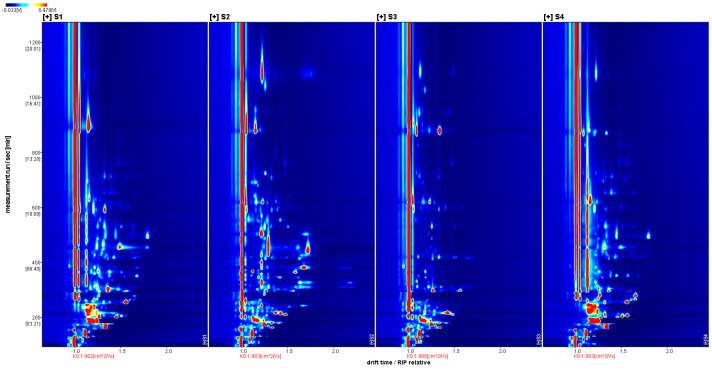
Two-dimensional GC-IMS spectra of volatile components in sausages.

**Figure 3 molecules-29-03772-f003:**
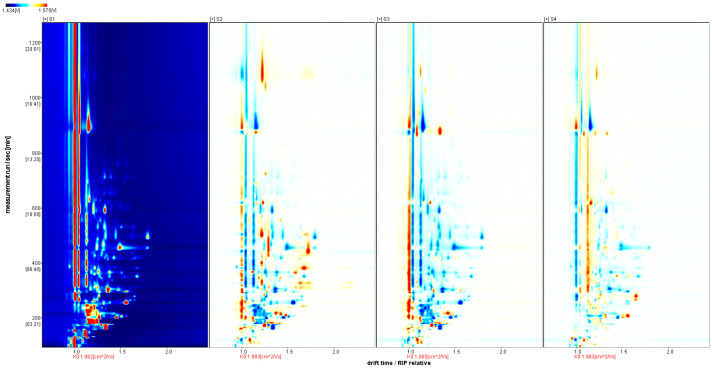
GC-IMS difference spectrum of volatile components in sausage.

**Figure 4 molecules-29-03772-f004:**
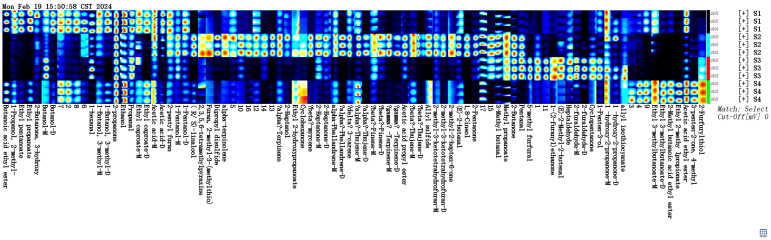
Fingerprints of volatile components in sausages.

**Figure 5 molecules-29-03772-f005:**
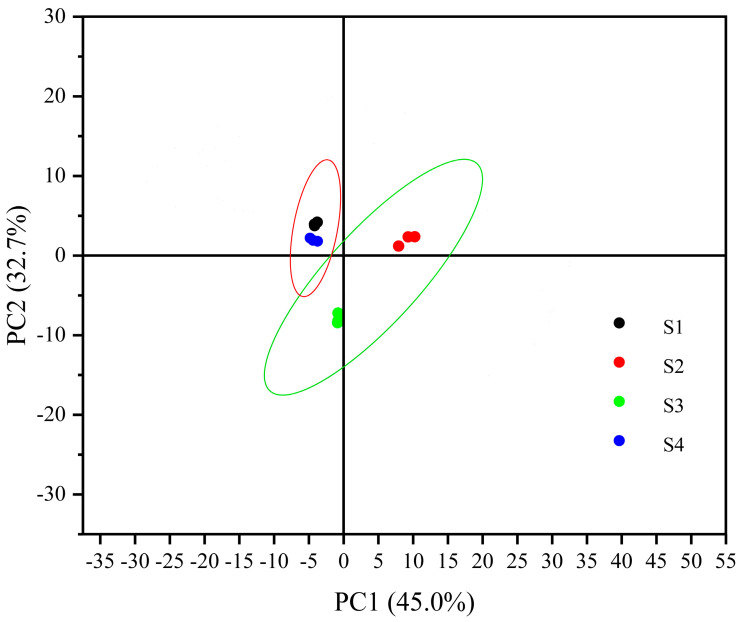
Plot of PCA scores of different sausage samples.

**Table 1 molecules-29-03772-t001:** Relative content of each class of volatile organic compounds in sausages.

Number of Individuals	Type	S1	S2	S3	S4
12	Hydrocarbons	4.66 ± 1.05 ^b^	24.09 ± 1.6 ^a^	3.75 ± 1.78 ^b^	6.58 ± 2.23 ^b^
11	Alcohols	21.69 ± 0.22 ^a^	11.67 ± 0.12 ^d^	16.92 ± 0.35 ^c^	18.92 ± 0.25 ^b^
10	Ketones	21.89 ± 0.24 ^d^	28.03 ± 0.2 ^b^	34.33 ± 0.85 ^a^	23.9 ± 0.17 ^c^
9	Aldehydes	5.88 ± 0.46 ^bc^	7.67 ± 0.96 ^b^	31.45 ± 1.31 ^a^	5.36 ± 1.55 ^c^
12	Esters	21.39 ± 0.4 ^b^	5.43 ± 0.82 ^d^	6.81 ± 0.78 ^c^	29.29 ± 0.67 ^a^
1	Acids	23.09 ± 0.34 ^a^	13.42 ± 2.92 ^b^	6.19 ± 1.04 ^c^	14.58 ± 1.17 ^b^
10	Others	1.81 ± 0.03 ^c^	10.45 ± 0.87 ^a^	1.79 ± 0.16 ^c^	2.84 ± 0.06 ^b^

Note: Different lowercase letters indicate significant differences between groups (*p* < 0.05).

**Table 2 molecules-29-03772-t002:** Volatile organic compounds in sausages.

Count	Compound	Type	CAS#	Formula	MW	RI	Rt [s]	S1	S2	S3	S4
1	Alpha-terpinolene	Hydrocarbons	C586629	C_10_H_16_	136.2	1276.9	570.964	0.05 ± 0.01 ^b^	0.2 ± 0.02 ^a^	0.02 ± 0 ^b^	0.03 ± 0 ^b^
2	Alpha-Terpinene	Hydrocarbons	C99865	C_10_H_16_	136.2	1191.2	427.41	0.33 ± 0.06 ^b^	0.97 ± 0.04 ^a^	0.19 ± 0.01 ^c^	0.31 ± 0 ^b^
3	Alpha-Phellandrene-M	Hydrocarbons	C99832	C_10_H_16_	136.2	1161.2	384.025	0.49 ± 0.09 ^b^	0.83 ± 0.05 ^a^	0.17 ± 0.01 ^d^	0.3 ± 0.01 ^c^
4	Beta-Thujene-M	Hydrocarbons	C28634891	C_10_H_16_	136.2	1119	330.28	0.18 ± 0.03 ^c^	1.36 ± 0.14 ^a^	0.21 ± 0.01 ^c^	0.67 ± 0.05 ^b^
5	Gamma -Terpinene-M	Hydrocarbons	C99854	C_10_H_16_	136.2	1238.7	501.954	0.1 ± 0.01 ^b^	4.3 ± 0.53 ^a^	0.31 ± 0 ^b^	0.23 ± 0.01 ^b^
6	Beta-myrcene	Hydrocarbons	C123353	C_10_H_16_	136.2	1174.1	402.085	0.09 ± 0 ^bc^	1.63 ± 0.04 ^a^	0.07 ± 0 ^c^	0.13 ± 0.01 ^b^
7	Alpha-Phellandrene-D	Hydrocarbons	C99832	C_10_H_16_	136.2	1159.6	381.806	0.22 ± 0.06 ^b^	3.87 ± 0.12 ^a^	0.08 ± 0 ^b^	0.12 ± 0.01 ^b^
8	Delta 3-carene	Hydrocarbons	C13466789	C_10_H_16_	136.2	1142.4	359.143	0.48 ± 0.07 ^b^	0.67 ± 0.04 ^a^	0.1 ± 0 ^d^	0.27 ± 0 ^c^
9	Beta-Thujene-D	Hydrocarbons	C28634891	C_10_H_16_	136.2	1117.8	328.845	0.12 ± 0.03 ^b^	2.21 ± 0.11 ^a^	0.17 ± 0.01 ^b^	0.27 ± 0.03 ^b^
10	Beta-Pinene-M	Hydrocarbons	C127913	C_10_H_16_	136.2	1104.2	313.338	0.2 ± 0.02 ^c^	0.24 ± 0.03 ^b^	0.36 ± 0 ^a^	0.11 ± 0.01 ^d^
11	Alpha-Thujene-M	Hydrocarbons	C2867052	C_10_H_16_	136.2	1025.1	243.986	1.71 ± 0.03 ^b^	1.57 ± 0.1 ^c^	0.51 ± 0.03 ^d^	2.27 ± 0.01 ^a^
12	Alpha-Thujene-D	Hydrocarbons	C2867052	C_10_H_16_	136.2	1026.3	244.899	0.05 ± 0.01 ^b^	0.69 ± 0.16 ^a^	0.07 ± 0.01 ^b^	0.04 ± 0 ^b^
13	Gamma -Terpinene-D	Hydrocarbons	C99854	C_10_H_16_	136.2	1240.6	505.13	0.16 ± 0 ^b^	1.06 ± 0.41 ^a^	0.13 ± 0.01 ^b^	0.19 ± 0.01 ^b^
14	Beta-Pinene-D	Hydrocarbons	C127913	C_10_H_16_	136.2	1103.3	312.283	0.14 ± 0.03 ^c^	1.04 ± 0.08 ^a^	0.34 ± 0 ^b^	0.09 ± 0.01 ^c^
15	5-methyl furfural	Aldehydes	C620020	C_6_H_6_O_2_	110.1	1557.3	1092.06	0.47 ± 0.02 ^c^	0.24 ± 0.02 ^c^	3.88 ± 0.24 ^a^	1.29 ± 0.09 ^b^
16	2-furaldehyde-M	Aldehydes	C98011	C_5_H_4_O_2_	96.1	1457.8	873.816	0.83 ± 0.05 ^b^	0.58 ± 0.06 ^b^	7.34 ± 0.48 ^a^	1.09 ± 0.64 ^b^
17	2-furaldehyde-D	Aldehydes	C98011	C_5_H_4_O_2_	96.1	1459.8	877.84	0.1 ± 0.02 ^b^	0.39 ± 0.33 ^b^	9.19 ± 0.54 ^a^	0.45 ± 0.48 ^b^
18	Heptaldehyde	Aldehydes	C111717	C_7_H_14_O	114.2	1186.6	420.523	0.22 ± 0.02 ^bc^	0.3 ± 0.07 ^b^	1.02 ± 0.08 ^a^	0.17 ± 0 ^c^
19	(E)-2-Methyl-2-butenal	Aldehydes	C497030	C_5_H_8_O	84.1	1104.2	313.331	0.07 ± 0 ^b^	0.11 ± 0.05 ^b^	0.55 ± 0.02 ^a^	0.09 ± 0 ^b^
20	1-hexanal	Aldehydes	C66251	C_6_H_12_O	100.2	1089.5	298.339	1.25 ± 0.22 ^b^	0.62 ± 0.01 ^d^	2.08 ± 0.21 ^a^	0.22 ± 0.01 ^c^
21	Propanal	Aldehydes	C123386	C_3_H_6_O	58.1	779.5	130.377	1.3 ± 0.12 ^b^	1.02 ± 0.05 ^c^	2.27 ± 0.04 ^a^	0.86 ± 0.01 ^d^
22	3-Methyl butanal	Aldehydes	C590863	C_5_H_10_O	86.1	910.4	180.27	0.21 ± 0.02 ^b^	0.62 ± 0.04 ^a^	0.68 ± 0.05 ^a^	0.13 ± 0.01 ^c^
23	Pentanal	Aldehydes	C110623	C_5_H_10_O	86.1	983	215.754	0.55 ± 0.02 ^c^	2.07 ± 0.08 ^b^	3.9 ± 0.05 ^a^	0.21 ± 0.01 ^d^
24	(E)-2-butenal	Aldehydes	C123739	C_4_H_6_O	70.1	1050.9	264.437	0.87 ± 0.02 ^a^	0.83 ± 0.1 ^a^	0.36 ± 0.03 ^c^	0.62 ± 0 ^b^
25	Linalool	Alcohols	C78706	C_10_H_18_O	154.3	1555.7	1087.981	0.36 ± 0.07 ^c^	11.75 ± 0.75 ^a^	0.62 ± 0.28 ^c^	4.32 ± 0.29 ^b^
26	2-Heptanol	Alcohols	C543497	C_7_H_16_O	116.2	1306.1	622.241	0.1 ± 0.02 ^b^	0.18 ± 0.06 ^a^	0.06 ± 0.01 ^b^	0.04 ± 0 ^b^
27	1-Pentanol-M	Alcohols	C71410	C_5_H_12_O	88.1	1255.8	531.728	0.72 ± 0.02 c	1.22 ± 0.09 ^b^	1.4 ± 0.02 ^a^	0.43 ± 0.01 ^d^
28	1-Pentanol-D	Alcohols	C71410	C_5_H_12_O	88.1	1256.2	532.43	0.39 ± 0.05 ^a^	0.29 ± 0.04 ^b^	0.2 ± 0.01 ^c^	0.19 ± 0.02 ^c^
29	1-Butanol, 3-methyl-M	Alcohols	C123513	C_5_H_12_O	88.1	1211.2	457.353	1.11 ± 0.1 ^a^	0.42 ± 0.01 ^c^	0.36 ± 0.03 ^c^	0.68 ± 0.01^b^
30	1-Butanol, 3-methyl-D	Alcohols	C123513	C_5_H_12_ O	88.1	1209.4	454.547	3.96 ± 0.06 ^a^	0.44 ± 0.02 ^c^	0.18 ± 0.05 ^d^	1.82 ± 0.05 ^b^
31	1-Penten-3-ol	Alcohols	C616251	C_5_H_10_O	86.1	1163.7	387.468	0.42 ± 0.03 ^b^	0.18 ± 0.01 ^c^	0.99 ± 0.04 ^a^	0.16 ± 0.01 ^c^
32	Butanol-M	Alcohols	C71363	C_4_H_10_O	74.1	1149.9	368.875	0.46 ± 0.05 ^b^	0.26 ± 0.01 ^c^	0.99 ± 0.02 ^a^	0.24 ± 0.01 ^c^
33	Butanol-D	Alcohols	C71363	C_4_H_10_O	74.1	1149.4	368.186	1.14 ± 0.05 ^a^	0.09 ± 0.01 ^d^	0.21 ± 0 ^c^	0.45 ± 0.01 ^b^
34	1-Propanol, 2-methyl-	Alcohols	C78831	C_4_H_10_O	74.1	1089.7	298.482	3.27 ± 0.16 ^a^	0.53 ± 0.08 ^c^	0.63 ± 0.07 ^c^	1.91 ± 0.05 ^b^
35	2-Furfurylthiol	Alcohols	C98022	C_5_H_6_OS	114.2	1434.7	829.928	0.01 ± 0 ^d^	0.02 ± 0 ^c^	0.07 ± 0 ^a^	0.05 ± 0.01 ^b^
36	Ethanol	Alcohols	C64175	C_2_H_6_O	46.1	934.5	191.359	10.03 ± 0.04 ^c^	6.67 ± 0.2 ^d^	11.72 ± 0.18 ^b^	12.12 ± 0.11 ^a^
37	1-hydroxy-2-propanone-M	Ketones	C116096	C_3_H_6_O_2_	74.1	1307.3	623.866	4.55 ± 0.11 ^a^	2.96 ± 0.56 ^b^	5.47 ± 0.58 ^a^	5.08 ± 0.11 ^a^
38	1-hydroxy-2-propanone-D	Ketones	C116096	C_3_H_6_O_2_	74.1	1305	620.698	1.01 ± 0.52 ^a^	0.95 ± 0.8 ^a^	1.92 ± 0.86 ^a^	0.71 ± 0.02 ^a^
39	2-Butanone, 3-hydroxy	Ketones	C513860	C_4_H_8_O_2_	88.1	1288.5	593.684	1.65 ± 0.62 ^a^	0.75 ± 0.38 ^ab^	1.03 ± 0.38 ^ab^	0.38 ± 0.04 ^b^
40	2-methyl-2-hepten-6-one	Ketones	C110930	C_8_H_14_O	126.2	1346.3	680.985	0.03 ± 0 ^bc^	0.12 ± 0.01 ^a^	0.02 ± 0.01 ^c^	0.04 ± 0 ^b^
41	Cyclohexanone	Ketones	C108941	C_6_H_10_O	98.1	1287	590.714	0.01 ± 0 ^b^	0.04 ± 0.01 ^a^	0.02 ± 0 ^b^	0.02 ± 0.01 ^b^
42	Cyclopentanone	Ketones	C120923	C_5_H_8_O	84.1	1188.1	422.686	0.07 ± 0.01 ^c^	0.08 ± 0 ^c^	0.35 ± 0.02 ^a^	0.14 ± 0 ^b^
43	2-Heptanone-M	Ketones	C110430	C_7_H_14_O	114.2	1182.5	414.308	0.22 ± 0.01 ^b^	0.72 ± 0.05 ^a^	0.78 ± 0.02 ^a^	0.16 ± 0.01 ^c^
44	2-Heptanone-D	Ketones	C110430	C_7_H_14_O	114.2	1183.9	416.427	0.25 ± 0.02 ^b^	0.53 ± 0.06 ^a^	0.29 ± 0.01 ^b^	0.12 ± 0.01 ^c^
45	2-propanone	Ketones	C67641	C_3_H_6_O	58.1	820.9	144.417	12.71 ± 0.07 ^c^	10.75 ± 0.46 ^d^	15.52 ± 0.25 ^a^	14.13 ± 0.04 ^b^
46	2-Pentanone	Ketones	C107879	C_5_H_10_O	86.1	988.3	218.614	0.44 ± 0.03 ^c^	2.32 ± 0.1 ^a^	1.48 ± 0.03 ^b^	0.2 ± 0.01 ^d^
47	2-Butanone	Ketones	C78933	C_4_H_8_O	72.1	898.3	174.929	0.85 ± 0.02 ^d^	5.46 ± 0.26 ^b^	7.19 ± 0.05 ^a^	1.28 ± 0.02 ^c^
48	3-penten-2-one, 4-methyl	Ketones	C141797	C_6_H_10_O	98.1	1138.2	353.703	0.02 ± 0 ^c^	0.06 ± 0.01 ^b^	0.05 ± 0.01 ^b^	0.61 ± 0.01 ^a^
49	Ethyl 2-hydroxypropanoate	Esters	C97643	C_5_H_10_O_3_	118.1	1321	643.29	0.08 ± 0.01 ^a^	0.08 ± 0.02 ^a^	0.04 ± 0.01 ^b^	0.09 ± 0 ^a^
50	Ethyl caproate-M	Esters	C123660	C_8_H_16_O_2_	144.2	1235.6	496.646	1.02 ± 0.08 ^a^	0.35 ± 0.31 ^a^	0.74 ± 0.64 ^a^	0.52 ± 0.01 ^a^
51	Ethyl caproate-D	Esters	C123660	C_8_H_16_O_2_	144.2	1234.7	495.242	1.35 ± 0.06 ^a^	0.06 ± 0 ^b^	0.06 ± 0.01 ^b^	1.43 ± 0.1 ^a^
52	Ethyl pentanoate	Esters	C539822	C_7_H_14_O_2_	130.2	1136	350.97	0.73 ± 0.04 ^a^	0.05 ± 0.01 ^c^	0.05 ± 0.01 ^c^	0.21 ± 0.01 ^b^
53	Ethyl 3-methylbutanoate-D	Esters	C108645	C_7_H_14_O_2_	130.2	1069.7	280.435	0.35 ± 0.03 ^b^	0.04 ± 0.01 ^c^	0.04 ± 0 ^c^	2.72 ± 0.08 ^a^
54	Ethyl 3-methylbutanoate-M	Esters	C108645	C_7_H_14_O_2_	130.2	1070.2	280.864	0.47 ± 0.01 ^b^	0.11 ± 0.01 ^c^	0.02 ± 0 ^d^	0.76 ± 0.01 ^a^
55	Butanoic acid ethyl ester	Esters	C105544	C_6_H_12_O_2_	116.2	1043.1	258.09	5.84 ± 0.24 ^a^	0.19 ± 0.09 ^b^	0.15 ± 0.04 ^b^	5.57 ± 0.22 ^a^
56	2-Methyl butanoic acid ethyl ester	Esters	C7452791	C_7_H_14_O_2_	130.2	1055.1	267.973	0.16 ± 0.01 ^b^	0.01 ± 0 ^c^	0.02 ± 0 ^c^	1 ± 0.03 ^a^
57	Ethyl 2-methy lpropionate	Esters	C97621	C_6_H_12_O_2_	116.2	968.6	208.208	0.18 ± 0.01 ^b^	0.04 ± 0.01 ^b^	0.07 ± 0 ^b^	3.44 ± 0.14 ^a^
58	Ethyl propanoate	Esters	C105373	C_5_H_10_O_2_	102.1	958.1	202.875	2.3 ± 0.08 ^a^	0.06 ± 0.03 ^c^	0.06 ± 0.03 ^c^	1.11 ± 0.08 ^b^
59	Acetic acid ethyl ester	Esters	C141786	C_4_H_8_O_2_	88.1	875.8	165.477	8.18 ± 0.05 ^b^	1.23 ± 0.07 ^d^	3.38 ± 0.04 ^c^	10.56 ± 0.07 ^a^
60	Acetic acid propyl ester	Esters	C109604	C_5_H_10_O_2_	102.1	980	214.14	0.44 ± 0.01 ^c^	1.93 ± 0.1 ^a^	1.28 ± 0.05 ^b^	0.36 ± 0.01 ^c^
61	Methyl propanoate	Esters	C554121	C_4_H_8_O_2_	88.1	913.7	181.748	0.15 ± 0.02 ^c^	0.55 ± 0.03 ^b^	0.64 ± 0.03 ^a^	0.14 ± 0 ^c^
62	Acetic acid-M	Acids	C64197	C_2_H_4_O_2_	60.1	1463.7	885.381	8.81 ± 0.14 ^a^	7.45 ± 0.67 ^b^	4.56 ± 0.65 ^c^	8.99 ± 0.51 ^a^
63	Acetic acid-D	Acids	C64197	C_2_H_4_O_2_	60.1	1462.3	882.662	14.19 ± 0.38 ^a^	4.41 ± 1.56 ^b^	1.59 ± 0.19 ^c^	4.96 ± 0.39 ^b^
64	2,3,5,6-tetramethylpyrazine	Others	C1124114	C_8_H_12_N_2_	136.2	1463.7	885.381	0.02 ± 0 ^a^	0.05 ± 0.01 ^a^	0.04 ± 0.01 ^a^	0.04 ± 0.03 ^a^
65	1-(2-furanyl)ethanone	Others	C1192627	C_6_H_6_O_2_	110.1	1493.7	946.925	0.03 ± 0 ^c^	0.02 ± 0 ^c^	0.39 ± 0.02 ^a^	0.14 ± 0.01 ^b^
66	Dipropyl disulfide	Others	C629196	C_6_H_14_S_2_	150.3	1376.8	728.814	0.07 ± 0 ^b^	0.68 ± 0.09 ^a^	0.1 ± 0.03 ^b^	0.11 ± 0.02 ^b^
67	Furan, 2-methyl-3-(methylthio)	Others	C63012975	C_6_H_8_OS	128.2	1354.5	693.503	0.02 ± 0 ^c^	0.05 ± 0.01 ^a^	0.02 ± 0 ^b^	0.01 ± 0 ^c^
68	2-pentyl furan	Others	C3777693	C_9_H_14_O	138.2	1230.4	487.946	0.36 ± 0.01 ^a^	0.24 ± 0.01 ^c^	0.28 ± 0.01 ^b^	0.21 ± 0.01 ^d^
69	2-methyl-3-ketotetrahydrofuran-M	Others	C3188009	C_5_H_8_O_2_	100.1	1269.2	556.292	0.44 ± 0.03 ^b^	0.96 ± 0.09 ^a^	0.5 ± 0.02 ^b^	0.49 ± 0.01 ^b^
70	2-methyl-3-ketotetrahydrofuran-D	Others	C3188009	C_5_H_8_O_2_	100.1	1269.4	556.692	0.06 ± 0 ^b^	0.46 ± 0.08 ^a^	0.06 ± 0.01 ^b^	0.07 ± 0 ^b^
71	Allyl sulfide	Others	C592881	C_6_H_10_S	114.2	1150.5	369.559	0.4 ± 0.02 ^b^	1.45 ± 0.09 ^a^	0.05 ± 0.01 ^c^	1.41 ± 0.05 ^a^
72	Allyl isothiocyanate	Others	C57067	C_4_H_5_NS	99.2	1374.6	725.258	0.07 ± 0.01^a^	0.08 ± 0.01 ^a^	0.24 ± 0.09 ^a^	0.13 ± 0.01 ^a^
73	1,8-Cineol	Others	C470826	C_10_H_18_O	154.3	1202.3	443.795	0.32 ± 0.01^b^	5.22 ± 0.3 ^a^	0.11 ± 0.01 ^b^	0.11 ± 0 ^b^

Note: Different lowercase letters indicate significant differences between groups (*p* < 0.05).

**Table 3 molecules-29-03772-t003:** ROAV values for VOCs in sausages.

Count	Compound	Odor Threshold (μg/kg)	S1	S2	S3	S4
1	Alpha-terpinolene	41	0.000279	0.000342	0.001094	5.36 × 10^−5^
2	Alpha-Terpinene	7.9	0.010507	0.008472	0.048965	0.002752
3	Alpha-Phellandrene-M	40	0.003054	0.001424	0.008615	0.000525
4	Beta-Thujene-M		——	——	——	——
5	Gamma-Terpinene-M	65	0.000387	0.00455	0.009829	0.000255
6	Beta-myrcene	1.2	0.018072	0.093502	0.119976	0.007849
7	Alpha-Phellandrene-D	40	0.0014	0.006658	0.004177	0.000211
8	Delta 3-carene	770	0.000156	5.99 × 10^−5^	0.000254	2.45 × 10^−5^
9	Beta-Pinene-M	60	0.00082	0.000279	0.012379	0.000128
10	Alpha-Thujene-M		——	——	——	——
11	Gamma -Terpinene-D	55	0.000712	0.001323	0.004794	0.000245
12	Beta-Pinene-D	60	0.000578	0.001187	0.011626	0.000111
	Hydrocarbons12					
13	5-methyl furfural	500	0.000235	3.28 × 10^−5^	0.015908	0.000184
14	2-furaldehyde-M	3000	7.74 × 10^−5^	2.22 × 10^−5^	0.011288	3.64 × 10^−5^
15	Heptaldehyde	2.8	0.019812	0.007371	0.744409	0.004278
16	(E)-2-Methyl-2-butenal	4.4	0.003907	0.001642	0.253965	0.001391
17	1-hexanal	4.5	0.06935	0.009533	0.945901	0.003513
18	Propanal	15.1	0.021486	0.00464	0.308171	0.004037
19	3-Methyl butanal	1.1	0.047273	0.038626	1.274982	0.008453
20	Pentanal	12	0.011531	0.011834	0.665159	0.001271
21	(E)-2-butenal	0.3	0.722664	0.189462	2.460309	0.145959
	Aldehydes 9					
22	Linalool	6	0.014953	0.13458	0.213268	0.051098
23	2-Heptanol	0.1	0.262801	0.125326	1.200566	0.031462
24	1-Pentanol-M	150.2	0.001855	0.000693	0.021836	0.000293
25	1-Butanol, 3-methyl-M	250	0.001113	0.000114	0.002972	0.000194
26	1-Butanol, 3-methyl-D	250	0.003968	0.00012	0.001495	0.000519
27	1-Penten-3-ol	358.1	0.000293	3.51 × 10^−5^	0.005681	3.2 × 10^−5^
28	Butanol-M	459.2	0.000253	3.93 × 10^−5^	0.004398	3.64 × 10^−5^
29	Butanol-D	459.2	0.00062	1.32 × 10^−5^	0.000948	7 × 10^−5^
30	1-Propanol, 2-methyl-		——	——	——	——
31	2-Furfurylthiol	0.008	0.188131	0.141054	17.01225	0.476383
32	Ethanol	950,000	2.64 × 10^−6^	4.83 × 10^−7^	2.53 × 10^−5^	9.06 × 10^−7^
	Alcohols 11					
33	1 -hydroxy-2-propanone-M	10,000	0.000164	3.34 × 10^−5^	0.00191	4.61 × 10^−5^
34	2-Butanone, 3-hydroxy	0.014	29.46011	3.694953	151.1137	1.905421
35	2-methyl-2-hepten-6-one	0.3	0.022844	0.027201	0.141979	0.009972
36	Cyclohexanone	5.27	0.000263	0.000546	0.007102	0.000231
37	Cyclopentanone	31–1120	——	——	——	——
38	2-Heptanone-M	140	0.000857	0.000614	0.015662	0.000138
39	2-propanone	40,000	7.96 × 10^−5^	1.85 × 10^−5^	0.000795	2.51 × 10^−5^
40	2-Pentanone	98	0.001131	0.00163	0.03084	0.000145
41	2-Butanone	35,400.2	6.03 × 10^−6^	1.06 × 10^−5^	0.000416	2.58 × 10^−6^
42	3-penten-2-one, 4-methyl	48	0.000112	8.18 × 10^−5^	0.001925	0.00091
	Ketones 10					
43	Ethyl 2-hydroxypropanoate	8	0.00242	0.00066	0.00922	0.000795
44	Ethyl caproate-M	1	0.593224	0.027999	1.63507	0.138246
45	Ethyl pentanoate	1.5	0.121761	0.002385	0.062388	0.009772
46	Ethyl 3-methylbutanoate-D	0.58	0.149325	0.004927	0.126278	0.33356
47	Ethyl 3-methylbutanoate-M	0.58	0.203934	0.012652	0.065263	0.092728
48	Butanoic acid ethyl ester	0.053	27.56826	0.252359	5.706915	7.471426
49	2-Methyl butanoic acid ethyl ester	0.968–11.7	——	——	——	——
50	Ethyl 2-methy lpropionate	0.1	0.450194	0.029475	1.450755	2.442442
51	Ethyl propanoate	0.027	21.28589	0.153615	4.698446	2.916468
52	Acetic acid ethyl ester	5	0.409878	0.016886	1.384313	0.149998
53	Acetic acid propyl ester	2000	5.51 × 10^−5^	6.63 × 10^−5^	0.001307	1.29 × 10^−5^
54	Methyl propanoate	100	0.000383	0.000377	0.013011	9.62 × 10^−5^
	Esters 12					
55	Acetic acid-M	22,000	0.000262	3.71 × 10^−5^	0.000573	4.5 × 10^−5^
	Acids 1					
56	2,3,5,6-tetramethylpyrazine	2.5	0.002429	0.001502	0.029068	0.001153
57	1-(2-furanyl) ethanone	10	0.00087	0.000134	0.080027	0.000995
58	Dipropyl disulfide	0.13	0.131401	0.357982	1.540802	0.061862
59	Furan, 2-methyl-3-(methylthio)	0.2	0.01894	0.018544	0.245471	0.005128
60	2-pentyl furan	6	0.015218	0.002758	0.097303	0.002428
61	2-methyl-3-ketotetrahydrofuran-M	6	0.018295	0.010998	0.169048	0.005853
62	2-methyl-3-ketotetrahydrofuran-D	6	0.002664	0.005234	0.019049	0.000824
63	Allyl sulfide	0.001	100	100	100	100
64	1,8-Cineol	15	0.005407	0.023932	0.01449	0.000504
65	Allyl isothiocyanate	0.19	0.097018	0.028778	2.588437	0.048537
	Others 10					

## Data Availability

The original contributions presented in the study are included in the article. Further inquiries can be directed to the corresponding authors.

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
