# Peer review of "Comparative Analysis of Commercially Available Flavor Oil Sausages and Smoked Sausages"

_molecules, 2024, doi:10.3390/molecules29163772_

Round 1

Reviewer 1 Report

Comments and Suggestions for Authors

Based on my review of your manuscript titled “Comparative Analysis of Commercially Available Flavor Oil Sausages and Smoked Sausages”, I find that the overall quality of the paper is commendable. The manuscript is generally well-written and organized.

The study aims to explore the variability in quality, taste, texture, color, and flavor among different types of sausages, focusing on comparing the influence of processing techniques and raw material preparation. It underscores the significance of these factors in enhancing sausage quality, with a specific emphasis on volatile organic compounds (VOCs) and texture as critical evaluation criteria. Notably, the manuscript discusses how sausage production methods like steaming and smoking contribute to aroma and texture through chemical reactions such as protein denaturation and fat oxidation.

Regarding Figures and Tables, while they are essential for conveying information, I recommend improving their labeling and clarity. Ensure that figure legends are descriptive, and tables include concise headings. Consider integrating some tables into the main text where they can enhance the coherence of your arguments.

In terms of Language and Style, the manuscript is generally well-written, but it could benefit from addressing some grammatical errors and awkward phrasings. A thorough proofreading byb an English professional would help improve clarity and overall coherence.

Overall, while the manuscript presents promising findings and insights, revisions are necessary to address the outlined concerns. With appropriate adjustments, particularly in restructuring sections for better flow, enhancing figure and table clarity, and refining language for precision, this study could make a significant contribution to the field. Therefore, I recommend accepting the manuscript with revisions.

Please address the following points in your revision to strengthen the manuscript for publication consideration.

General Comments:

1. there are some places in the text where there is ambiguous language that will confuse the reader, such as 3. in the conclusion section “smoked and flavor Oil Sausages are generally preferred by consumers over flavor Oil Sausages”, please check and revise.

2.Abstract should briefly summarize the objectives, methodology, major findings and implications of the study, please reorganize.

3. The P in P < 0.05 should be italicized, please check the whole text and revise.

3. Some references do not give specific page numbers, please revise according to the journal requirements.

4. 4 sausages are listed in the text, what is the basis of selection?

5. Figures and tables are essential but require better labeling and clarity. Figure legends should be descriptive, and tables should include concise headings. Consider integrating some tables into the main text where relevant.

6. The manuscript is generally well-written, but grammatical errors and awkward phrasings need correction. Consider a thorough proofreading for clarity and coherence.

 7. BTW, is “Flavor Oil Sausages” a correct term?

Comments on the Quality of English Language

should be revised by an English professional.

Author Response

Thank you for your valuable and professional feedback. The response content has been uploaded as an attachment. Please review it, and feel free to contact me if you have any questions. Once again, thank you for your input.

Reviewer 2 Report

Comments and Suggestions for Authors

In this review manuscript, the author applied gas chromatography-ion mobility spectroscopy (GC-IMS) to study the volatile flavor compounds in 4 types commercially available sausages. Also, comparative assessments of these 4 sausages were conducted by integrating sensory evaluation with textural and physicochemical parameters. Through fingerprinting combined principal component analysis (PCA) analysis and combined textural and sensory evaluations, the author draws the conclusion that smoked sausages are more favored by consumers than flavor oil sausages. However, this manuscript is not recommended for acceptance for the following reasons:

1.     Line 71, “widely used in the food industry” is not an intact sentence. Line 490, In Figure. 4 Fingerprints of volatile components in sausages, the contents are barely to read both horizontally and vertically.

2.     Line 108, section 2.3.3 the major experiment design - Measurement of volatile components didn’t provide enough details about the IMS setup and background info about quantitative analysis.

3.     The major research finding that 73 volatile compounds were detected in the four samples, and this finding indicates the flavor difference. However, there is no clear explanation about how this flavor difference contributes to the conclusion that smoked sausage more favored by consumers than oil sausages.  

4.     Only 4 sausages were tested, this small sample size has statistical limitation to represent the whole quality of smoked sausage and oil sausage.

Author Response

(The authors gave the same response as above.)

Round 2

Reviewer 2 Report

Comments and Suggestions for Authors

Thanks for answering all my questions. I have no more comments at this time. 

Author Response

Thank you for all your professional suggestions and comments on the manuscript, your supervision and evaluation of our manuscript quality has been improved, I wish you a happy life, work well!